# Comparative Study of the U(VI) Adsorption by Hybrid Silica-Hyperbranched Poly(ethylene imine) Nanoparticles and Xerogels

**DOI:** 10.3390/nano13111794

**Published:** 2023-06-02

**Authors:** Michael Arkas, Konstantinos Giannakopoulos, Evangelos P. Favvas, Sergios Papageorgiou, George V. Theodorakopoulos, Artemis Giannoulatou, Michail Vardavoulias, Dimitrios A. Giannakoudakis, Konstantinos S. Triantafyllidis, Efthalia Georgiou, Ioannis Pashalidis

**Affiliations:** 1National Centre for Scientific Research “Demokritos”, Institute of Nanoscience and Nanotechnology, 15310 Athens, Greecee.favvas@inn.demokritos.gr (E.P.F.); s.papageorgiou@inn.demokritos.gr (S.P.); g.theodorakopoulos@inn.demokritos.gr (G.V.T.); a.giannoulatou@acg.edu (A.G.); 2PYROGENESIS S.A.,19500 Attica, Greece; mvardavoulias@pyrogenesis-sa.gr; 3Department Chemistry, Aristotle University of Thessaloniki, 54124 Thessaloniki, Greece; dagchem@gmail.com (D.A.G.); ktrianta@chem.auth.gr (K.S.T.); 4Department of Chemistry, University of Cyprus, P.O. Box 20537, 1678 Nicosia, Cyprus; georgiou.efthalia@ucy.ac.cy

**Keywords:** uranyl cations, dendritic polymers, silica xerogels, composites, nanoparticles, water purification, radioactive wastewater, uranium removal, dendrimers, biomimetic

## Abstract

Two different silica conformations (xerogels and nanoparticles), both formed by the mediation of dendritic poly (ethylene imine), were tested at low pHs for problematic uranyl cation sorption. The effect of crucial factors, i.e., temperature, electrostatic forces, adsorbent composition, accessibility of the pollutant to the dendritic cavities, and MW of the organic matrix, was investigated to determine the optimum formulation for water purification under these conditions. This was attained with the aid of UV-visible and FTIR spectroscopy, dynamic light scattering (DLS), ζ-potential, liquid nitrogen (LN_2_) porosimetry, thermogravimetric analysis (TGA), and scanning electron microscopy (SEM). Results highlighted that both adsorbents have extraordinary sorption capacities. Xerogels are cost-effective since they approximate the performance of nanoparticles with much less organic content. Both adsorbents could be used in the form of dispersions. The xerogels, though, are more practicable materials since they may penetrate the pores of a metal or ceramic solid substrate in the form of a precursor gel-forming solution, producing composite purification devices.

## 1. Introduction

Nowadays, toxic metal radionuclide accumulation, particularly uranium, is rapidly accelerated due to the enormous technological development that continuously strives to satisfy military needs and increasing demands for energy [1,2]. Uranium exhibits a very complex behavior in water, comprising a wide spectrum of transitions between reduced or oxidized species, hydrolyzed forms, and complexes (Figure 1). Furthermore, precipitate formation, colloid aggregation, and sorption generate a multitude of chemical states that cause different side effects in the hydrosphere [1,2,3].

Adsorption proved the most cost-effective technique [4,5] for addressing the radioactivity of industrial wastewater and the decontamination of uranium-polluted aquatic systems among the other researched treatment methods (precipitation, solvent extraction, and ion exchange). It is mainly based on the complexation of the predominant form of uranium in water, i.e., the hexavalent uranyl cation (UO_2_^2+^), with the external groups of the adsorbing material (e.g., -NH_2_, -COOH, -OH), producing inner-sphere complexes, and the electrostatic interactions between the charged substrate surface and the opposite ionic species, affording outer-sphere analogues [6,7,8,9,10]. Intensive investigations performed recently led to the discovery of exceptional adsorbing substrates distinguished for both their specificity and effectiveness. The field extends from purely inorganic compounds such as carbon allotropes [11], metal oxides, and minerals [12] to organic-inorganic hybrids [11,13,14], composites [15,16], metal-organic frameworks (MOFs) [17,18,19], and aerogels [20,21,22]. Organic counterparts include conventional polymers [23,24], biopolymers [25,26], and dendritic polymers [27,28].

Hyperbranched polymers represent one of the three major categories of dendritic polymers [29,30,31,32,33,34,35]. Their polydispersity, inherited from their asymmetry, is the major difference from the symmetrical dendrimers [36,37,38,39,40]. Dendrons [41,42], on the other hand, are fragments of dendrimers or hyperbranched polymers that emanate from a central focal point. Together with the more exotic variants of dendrigrafts [43] and dendronized polymers [44,45,46,47], they form the family of radially polymerized dendritic polymers [48,49,50]: the fourth class of macromolecular architecture next to the linear, the branched, and the cross-linked [51].

Due to their particular irregular architecture, hyperbranched polymers resemble more closely the branched structures commonly encountered in nature. Some examples are the river deltas, the roots of the trees, the arteries of the blood circulatory system, the veins of the leaves, the fractals, and the dendritic cells [52]. The properties of hyperbranched polymers depend on their three main structural parts. The core or central focal point is usually formed by a similar type of monomer or by a functional moiety designed to introduce a specific functionality. The dense network of branches that generate internal cavities capable of hosting a wide variety of guest molecules or aggregates and the external-terminal groups that are responsible for the organization, solubility, and reactivity of the macromolecules frequently lead to their functionalization. The hyperbranched polymers share the same range of applications with their other dendritic polymers, for instance, in liquid crystals [53,54,55], drug delivery [56,57,58,59], tissue engineering [60], antimicrobial protection [61,62,63,64,65,66,67], diagnostics [68,69,70,71,72,73], gene transfection [74,75,76], and biosensors [77,78]. They have made a substantial contribution to the therapy of diseases such as cancer [79,80], conditions of the central nervous system [81,82], and rheumatoid arthritis [83,84]. In some of the most important implementations, the hyperbranched polymers are combined with inorganic substrates [85]. Examples of this category include (photo)catalysts [86,87,88], implants [89], and most notably, water purification [90,91,92]. In this context, our group has developed biomimetic silica-hyperbranched poly(ethylene imine) (PEI) nanospheres that proved capable of adsorbing a variety of pollutants [93]. By fine-tuning the inorganic/organic content ratio, this hybrid material was further evolved into gels that could be dried into xerogels. Gel formation optimally takes place in the pores of porous substrates, in particular ceramic substrates. Apart from dispersions, they are thus also capable of forming coatings and, in consequence, composite filters for continuous filtration. The scope of this work is to compare the two variants, i.e., nanospheres with higher content in hyperbranched nanosponge, against the more flexible and applicable xerogels under the demanding conditions required for the adsorption of U(IV) cation.

## 2. Materials and Methods

Hyperbranched poly(ethylene imines) PEIs (Mn = 2000, 5000, 25,000, and 750,000) were purchased from BASF, Ludwigshafen, Germany. Tetraethoxysilane, UO_2_(NO_3_) _2_‧6H_2_O, and P_2_O_5_ were obtained from Sigma-Aldrich, Steinheim, Germany; K_2_HPO_4_ from Carlo Erba Reagenti, Rodano, Milano, Italy; KH_2_PO_4_ from Merck, Darmstadt, Germany; and CoSO_4_·7H_2_O from Alpha Aesar (Ward Hill, MA, USA). All compounds were used as received.

The existence of solubilized adsorbents was determined by UV-vis spectroscopy employing a Cary 100 spectrophotometer Varian Inc., Palo Alto, CA, USA. The samples before and after U(VI) adsorption were characterized by FTIR (KBr) spectroscopy (using an FTIR spectrometer 8900, Shimadzu Duisburg, Germany) after drying the samples overnight in a vacuum oven at 70 °C. The morphology of the nanospheres and the xerogels was observed by scanning electron microscopy (SEM) using an FEI Quanta Inspect instrument FEI Hillsboro, OR, USA, which is also capable of Energy Dispersive X-Ray Spectroscopy (SEM-EDS) analysis. A thermal analysis study was conducted on a Setaram SETSYS Evolution 16/18 TGA/DSC analyzer, Caluire, France. The samples were heated up to 700 °C at a rate of 10 °C/min and remained at the terminal temperature for 3 h. The size of the aggregates at low pH before and after U(IV) sorption was determined in typical DLS experiments with the aid of an AXIOS-150/EX (Triton Hellas Thessaloniki Greece) with a He–Ne laser emitting at 658 nm and an Avalance detector at 90°. Electrostatic interactions were monitored by ζ-potential measurements on a ZetaPlus (Brookhaven Instruments Corporation Holtsville, NY, USA); the results represent the mean values of 10 measurements collected for each dispersion. N_2_ adsorption experiments at 77 K were performed on an Autosorb-1 gas analyzer with a Krypton upgrade (Quantachrome Corp. Boynton Beach, FL, USA). Before the isotherm measurements, the samples were outgassed overnight at 120 °C. The surface areas were calculated according to the Brunauer-Emmett-Teller (BET) equation, while the pore size distribution (PSD) was determined by the Non-Local Density Functional Theory (NLDFT) method.

### 2.1. Preparation of Hybrid Silica-PEI Nanoparticles

The PEI core-silica shell composites were synthesized according to the standard method disclosed by Knecht et al. [94,95] for poly(amidoamine) PAMAM and poly(propylene imine) PPI dendrimers, adapted for the non-symmetric PEI counterpart. Briefly, 10 mL of 20 mM (concerning the total of primary and secondary amine groups) hyperbranched PEI solutions (for all 4 different Mns) were buffered at pH 7.5 by phosphates. Silicic acid (1 M) was prepared from the hydrolysis of tetramethyl orthosilicate in 5 mM HCl under intense stirring for 15 min. Silica precipitation was instantaneous after adding 1 mL to the dendritic polymer solution. The product was isolated by centrifugation (10 min, 12,000× *g*) and washed twice with water (yield: 77%).

### 2.2. Preparation of Silica-PEI Xerogels

Acid hydrolysis of 5 mL tetraethoxysilane solution (1 M) with 25 µL HCl (1 M) under stirring for 15 min afforded 1 M orthosilicic acid as above. Afterwards, 5 mL of PEI solutions (40 mM concerning the total of primary and secondary amine groups) were added, and the pH was adjusted to 7.5 with KHPO_4_. Following 2 h, a hydrogel was formed and submitted to drying overnight under vacuum and over phosphorus pentoxide (P_2_O_5_) (Sigma-Aldrich, Steinheim, Germany) to form a silica/PEI xerogel.

### 2.3. Determination of Sorption Kinetics

Adsorption experiments of UO_2_^2+^ on the composite nanoparticles and xerogel samples were conducted as described elsewhere [22,96]. Briefly, the kinetic experiments were carried out in aqueous solutions (25 mL) containing 0.01 g (0.4 g/L) of each adsorbate and 0.01 mol/L UO_2_(NO_3_) _2_‧6H_2_O. The cation adsorption was investigated under ambient conditions. The spectrophotometric method was calibrated before and after each experiment using reference solutions, which were prepared by dissolving analytical grade UO_2_(NO_3_) _2_‧6H_2_O in de-ionized water under similar conditions.

### 2.4. Determination of Sorption Isotherms

Samples of varying concentrations of U(VI) at pH 3.0 or 4.0 were prepared. The initial U(VI) concentration was varied between 1 × 10^−6^ mol/L and 0.1 mol/L, the contact time was set at 24 h, and the determination of UO_2_^2+^ in the solutions was carried out by UV-Vis spectrophotometry directly at higher concentrations and using Arsenazo-III at lower U(VI) concentrations [22,96]. The pH values in the resulting diagrams correspond to the pH values determined at equilibrium after adsorption. There was no buffer needed to stabilize the pH because the desired pH was adjusted using 0.1 M HCl or 0.1 M NaOH and remained stable (pH 3 and pH 4) after adjustment. The effect of temperature was investigated at 298 K, 308 K, and 318 K at a U(VI) concentration of 5 × 10^−5^ mol/L.

The isothermal data were fitted with the Langmuir (Equation (1)) and Freundlich (Equation (2)) isotherm models.
(1)$$q_{e}=\frac{q_{\max}{\;K}_{L}{\;C}_{e}}{{1+K}_{L}{\;C}_{e}}$$
(2)$$q_{e}{=K}_{F}{\;C}_{e}{}^{1/n}$$
where q_e_ is the U(VI) uptake (in mol/kg) at equilibrium, C_e_ is the U(VI) concentration in solution at equilibrium (in mol/L), q_max_ is the adsorption capacity (in mol/kg), K_L_ is the Langmuir equilibrium constant (in L/mol), K_F_ is the Freundlich constant, and n is the empirical adsorption intensity of the Freundlich model.

The amount of U(VI) adsorbed at time t, q_t_ (mol kg^−1^), was calculated using Equation (3), where C_i_ (mol/L) is the initial U(VI) concentration in the solution, C_t_ (mol/L) is the final U(VI) concentration in the solution at time t, V (L) is the solution volume, and m (g) is the weight of xerogel and nanoparticle samples.
(3)$$q_{t}=\frac{\left(C_{i}-C_{t}\right)V}{m}$$

The experiments were performed in duplicate, and the mean values of the results have been used for data evaluation. The relative uncertainty of the values was below 10%.

## 3. Results and Discussion

### 3.1. Xerogel and Nanoparticle Characterization

#### 3.1.1. Thermogravimetry

From the experimental section, it can be easily deduced that in the case of nanoparticles, a fivefold quantity of hyperbranched poly(ethylene imine) is used concerning the xerogels. A first assessment of the final composition of the adsorbing materials is obtained with thermogravimetry and by comparison of the thermal decomposition profiles of the dendritic matrices. As can be seen in Figure 2a, PEI’s thermal decomposition profile contains roughly two major steps. The first up to 170 °C corresponds to the loss of humidity and potentially hydrogen-bonded ethanol deriving from the hydrolysis of tetraethoxysilane. The second depicts their total decomposition at 400 °C. The latter is completed with a considerable thermal hysteresis when the dendritic polymers are coupled with silica (Figure 2b). From the percentages of organic content that derive from the deduction of the moisture-alcoholic fraction from the total weight loss percentages (Table 1), it can be concluded that the organic content of the hybrid nanoparticles is more than double.

#### 3.1.2. IR Spectroscopy of Nanoparticles and Xerogels

Infrared spectra reveal more information about composite nanoparticles and xerogels. Table 2 summarizes the assignment of the observed absorption bands along with PEI 25,000 for comparison purposes. The typical amorphous silica vibrations are present [98], whereas the majority of the PEI bands are weak and difficult to detect. Some of them appear as shoulders, for instance, the antisymmetric C-N stretching band at 1105 cm^−1^ that is next to the broad Si–O–Si stretching at 1042 cm^−1^ for the nanoparticles and 1074 cm^−1^ for the xerogels, while some others, such as the respective symmetric C-N stretching vibration at 1045 cm^−1^ and the CH_2_ rocking vibrations at 760 cm^−1^, are completely overlapped. The latter is under the Si–O–Si bending band at 784 cm^−1^. All these observations are consistent with the typical spectra of silicas deriving from both biomimetic [94,99,100] and biogenic biosilicification [101,102,103]. The reason for the lower absorption intensities of the organic matrices resides in their encapsulation into a ceramic silica entourage. From the few discerned, the N–H asymmetric stretching vibration appears at 3400–3370 cm^−1^ and is partially overlapped by the broad absorption of the hydrogen-bonded Si-OH groups that are not transformed into siloxanes (3250 cm^−1^). Additionally, both symmetric and asymmetric stretchings of CH_2_ are present at 2981 and 2820 cm^−1^ for nanoparticles and 2962 and 2853 cm^−1^ for the xerogels [104]. They exhibit a shift to higher wavelengths in comparison to the respective PEI 25,000 bands (2935 and 2810 cm^−1^, respectively) due to the protonation of the PEI primary and secondary amine groups that balance the negative change of the Si–O^−^ groups (stretching band at 964 cm^−1^) of the developing silica gels and the forming nanoparticles. An analogous shift is encountered in the FTIR spectra of PEI–H_2_SO_4_ complexes [105]. This formation of ammonium groups additionally affects the N-H stretching, mainly the asymmetric [106]. The presence of the ammonium groups is further established by two new absorption bands at 1524 and 1473 cm^−1^ for the nanoparticles and 1530 cm^−1^ for the xerogels that correspond to the NH_x_^+^ symmetric and asymmetric bending of the Si–O^−^···NH_x_^+^ associations [107,108]. The characteristic Si-O-Si rocking band (540 cm^−1^) is also present [109], while a small sharp band observed at 3750 cm^−1^ in the case of xerogels is attributed to the SiO-H stretching of the non-hydrogen-bonded silanol groups.

#### 3.1.3. LN_2_ Porosimetry

N_2_ adsorption isotherms were performed to determine the structural characteristics of the developed samples, such as pore size distribution, total pore volume, and specific surface area. Through the nitrogen adsorption isotherms at 77 K, the adsorbed volumes were obtained (Figure 3a,b), with the BET (Brunauer-Emmett-Teller) surface areas [110] being between approximately 80 and 715 m^2^/g (Table 3). The “xerogel 25,000” initially exhibited a wider hysteresis loop and a higher pore distribution, and the specific area value was 242.0 m^2^/g. Exhaustive drying of this particular sample under vacuum led to a dramatic increase in the specific area to 715 m^2^/g and a “normalization” of the overall behavior of the sample. Drying conditions thus play a critical role in the porosity of the xerogels and trigger interest in further research. These specific areas are sufficiently comparable to those previously encountered in the literature for similar materials (Table 4). The measured total pore volumes fluctuated between 0.30 and 0.83 cc/g and 0.33 and 0.88 cc/g for xerogel and nanoparticle samples, respectively.

For the pore size distribution, the BJH (Barret, Joyner, and Halenda) [116] and NLDFT models were applied for the LN_2_ isotherm desorption branch. The pore sizes were determined to be between 2 and 80 nm for all studied samples. As observed (Figure 3a,b), each material’s group presents a different type of N_2_ adsorption isotherm. The xerogel samples present similar isotherms (Figure 3a) of type IV_(a)_ with a hysteresis loop [117]. The existence of these reproducible, permanent hysteresis loops is generally associated with capillary condensation and, in this case, can be attributed to network effects [117]. The main differences among the four isotherms lie in the different total adsorbed amounts as well as the shapes of the hysteresis loops. In specific, the xerogel samples with the lower molecular weights (2000, 5000, and 25,000) present type H2(b) hysteresis loops. This type of hysteresis is associated with pore blocking, and the size distribution of neck widths is larger compared to type H2(a). However, slight differences are presented between the xerogel samples 2000/25,000 and 5000. Indeed, from Figure 3c, it could be observed that the PSD curves of “xerogel 2000” and “xerogel 25,000” present a bimodal form (“xerogel 25,000” has a wider curve) compared to the narrow unimodal shape of the “xerogel 5000” sample. In addition, for the case of “xerogel 25,000”, with the highest outstanding surface area and the largest pore volume, its wider PSD curve is presented at the mesoporous area of 2–8 nm. Simultaneously, the sample possesses a slight amount of micropores (1.9 nm). Finally, the “xerogel 750,000” sample presents a type H5 hysteresis loop with a wider PSD curve (Figure 3c) from 4–10 nm mesopores containing both open and partially blocked mesopores. At this point, it could also be assumed that cavitation might be involved in the mechanism of desorption of this sample while the pore neck remains filled. Cavitation is broadly defined as the spontaneous formation and activity of bubbles in metastable liquids [118].

With the exception of the “xerogel 25,000”, which presents exceptional surface area and pore volume, the molecular weight seems to be a crucial parameter for the other xerogel samples, which determine the total pore volume and the surface area but not the formatted pore size, which remains almost the same. On the other hand, the nanoparticle samples present lower surface areas but higher total pore volumes and pore sizes (Table 3). The N_2_ adsorption isotherms (Figure 3b) are similar to type II with a sharp uptake at the higher relative pressures after P/P_0_ > 0.9, which forms a hysteresis loop during the desorption branch. According to the Kelvin equation, this is characteristic of materials containing large mesopores and macropores, as smoothly depicted in Figure 3d. Truly, a complex pore structure is disclosed with the existence of a broad range of pore sizes (pores from 6 up to 80 nm). Moreover, a pronounced difference concerning the adsorbed micropore volume of all samples is illustrated in the low-pressure region of the N_2_ (77 K) adsorption isotherms (insets in Figure 3a,b). Specifically, the xerogel samples present higher micropore volumes (“xerogel 25,000” approaches the maximum amount of 0.253 mL/g) compared to the other samples. On the other hand, the pore sizes of the nanoparticle samples (Figure 3c,d) are significantly larger than those of the xerogel samples, but overall their specific surface area is considerably smaller. It could be concluded that the observed differences between both variants of the developed materials reflect the different process steps (different precursor concentrations, drying conditions) involved during their preparation. An additional extensive study about the pore characteristics obtained from these hybrid materials correlated to their preparation parameters is in progress to shed light on their structural and textural properties.

#### 3.1.4. Size (DLS) and Charge (ζ-Potential)

Both the size of the pure adsorbents and the charge that develops at the interface between the surface of their solvation sphere and the liquid were measured at pH 3 obtained by nitric acid to simulate the conditions of the uranyl cation adsorption. Dynamic light scattering measurements of the PEI 5000 nanoparticles (Figure 4a) revealed aggregates with much bigger hydrodynamic radii (605 nm) than those measured in phosphate buffer pH 7 (384 nm) [93], indicating the development of bulkier solvation spheres around them, most probably due to the larger concentration of the nitrate anions. The nanoparticle size increases with increasing Mw of hyperbranched PEI. On the other hand, the particle’s charge decreases at the bulkier solvation spheres (Figure 4b), reflecting simply the fact that the same concentration of ammonium cations is distributed in a larger volume [119,120]. In contrast to the hybrid silica nanoparticles, the size and charge of the xerogel aggregates remain independent of PEI Mw. In this case, there are no separate silica shell polymer core entities, but dendritic “islands” dispersed in a continuous “sea” of silica.

### 3.2. Adsorption Kinetics

Silica is a well-known heavy metal adsorbent from water, and this property has been correlated with the electrostatic attractions between the positively charged metal ions and the negatively charged external silanol groups [121,122]. In this case, at pH 3, all adsorbing composites are positively charged, and electrostatic interactions are unfavorable. Uranium removal is attributed mainly to the presence of the dendritic polymer and the formation of inner-sphere complexes. This behavior is analogous to that of negatively charged dichromates and simple and hybrid silica nanoparticles [93]. Generally, the nanoparticles adsorb faster and contain about twice as much uranium as xerogels, while equilibrium for the former is established in about 2 h (Figure 5), a bit later than what was observed with lead and dichromate ions and at about the same time as mercury and cadmium cations [93].

### 3.3. Adsorption Isotherms

To evaluate the adsorption capacity of the dendrimers for the hexavalent uranium cations (UO_2_^2+^), batch-type adsorption experiments have been performed at pH 4.0 and pH 3.0 under ambient conditions. The experiments were carried out in the acidic pH range (pH 3.0 and pH 4.0) to avoid U(VI) solid phase precipitation due to the relatively high U(VI) concentrations used in the experiments (5 × 10^−6^ mol/L < [U(VI)] < 0.1 mol/L) [6,7,8,122]. The adsorption isotherms obtained at pH 4 are shown in Figure 6a and indicate that for the same size PEIs, the nanoparticles present higher adsorption capacities than xerogels, as expected due to their higher dendritic polymer content. Additionally, the molecular weight affects the adsorption efficiency. Except for PEI 5000 nanoparticles, which show the second-best adsorption efficiency, matrices of larger polymers result in better performance. This exception may be correlated with PEI 5000’s abnormally high BET surface area. The highest adsorption efficiency is observed for PEI 750,000 nanoparticles and the lowest for PEI 2000 counterparts, in conformity with the general tendency of materials with larger BET surface areas to exhibit higher pollutant retention potential.

Even at a 0.001 M U(VI) concentration in solution, the adsorption data did not reach a plateau, indicating extremely high adsorption capacities of the dendrimers for U(VI). To avoid U(VI) surface precipitation, which could interfere with the U(VI) sorption by the dendrimers, the adsorption experiments have also been performed at pH 3 using the PEI 750,000 nanoparticle and xerogel samples. The corresponding data, which are presented in Figure 6b, show that even at 0.01 M U(VI) concentration in solution, the adsorption data do not reach a plateau assuming extraordinary adsorption capacities (q_max_ > 7 mol/kg), which are supreme sorption capacity values, comparable to the best reported for aerogel materials [123] and superior to those of dendritic fibrous nano-silica bearing hyperbranched poly(amidoamine) (Table 5) [124].

#### Effects of Temperature and Calcination

The effect of temperature has been studied at three different temperatures (e.g., 25, 35, and 45 °C) for the U(VI) adsorption by the PEI 750,000 of nanoparticles (Figure 7a) and xerogels (Figure 7b). From the data summarized therein, it is obvious that for both material types, the adsorption efficiency increases significantly when the temperature increases from 25 °C to 35 °C, assuming that the adsorption is an endothermic, entropy-driven process. At 45 °C, the adsorption efficiency decreases, which is in contradiction with the previous statement. However, this could be attributed to an observed partial dissolution of the solid phase at 45 °C, which could explain the decline in the adsorption efficiency. The overall behavior of the U(VI) sorption by the hybrid silica dendritic polymer composites differs from what is generally observed for the adsorption of U(VI) by aerogel materials [123], dendritic fibrous nano-silica bearing hyperbranched poly(amidoamine) [124], and oxidized biochar fibers [6,7,8,9], which was found to be a purely endothermic, entropy-driven process.

In order to further investigate this discrepancy and inquire about the macroscopic material loss, 100 mg of the two samples (xerogel nanoparticles) were left under constant stirring at 45 °C for a week. Following centrifugation, we measured the supernatant’s UV-Vis absorbance. A broad peak at 230–200 nm (Figure 7c) could be attributed to PEI but is not characteristic. To verify PEI elusion into aqueous solutions at 45 °C, we attempted complexation with Co(II). The formation of a yellow complex permitted easy detection of PEI up to 0.2 mM (Figure 7c) (in primary and secondary amines), 200 times less than the initial concentration in the xerogels. Nevertheless, no such absorbance was observed in either supernatant. A possible explanation for the above-mentioned weight loss could be the release of the ethoxy silane hydrolysis byproduct, ethanol, which has a similar absorbance spectrum (Figure 7c). Probably some ethanol molecules were firmly incorporated in the amine-silanol hydrogen bond network and did not evaporate during drying. Thus, U(VI) adsorption to silica xerogels and nanoparticles is still endothermic, and the decrease in capacity at 45 °C is due to material dissolution that disrupts the pore structure of the adsorbent. Finally, elimination of the organic matrix by 3-h protracted heating at 700 °C reduces pollutant retention, but this decrease is not as dramatic as proven for the PEI 25,000 nanoparticles (Figure 7d).

### 3.4. Adsorbent Characterization Posterior to the Uranyl Adsorption

#### 3.4.1. Scanning Electron Microscopy (SEM)

Figure 8a,b depicts SEM micrographs of the hybrid nanoparticles and xerogel adsorbents, respectively, and provide a first perception of their morphology. The only difference between silica samples obtained through precipitation and gel-drying is the quasi-spherical structures that are present in Figure 8a, although to a lesser extent than previously recorded [97,131]. Uranium cubic crystals or other accumulation forms were not detected, in contrast to the previous observations on uranyl adsorption on polyamide microplastics [132]. The formation of SiO_2_ was established by energy-dispersive X-ray spectroscopy (Figure 8c for the nanoparticles, Figure 8d for the xerogels). The oxygen to silicon atomic ratio is about the same for the two samples (1.32 nanoparticles and 1.36 xerogels); however, the weight percentage of carbon is much larger in the nanoparticles (22.70 against 16.46), indicating the larger content of the dendritic matrix as registered in the thermogravimetry profiles. A slight increase in the uranium weight percentage in the nanoparticles corroborates the isotherm adsorption data on their better adsorption capacity. The concentration of the electron beam on the bright spots did not cause an increase in the uranium percentage (Figure 8c, gray profile), verifying that they originate from silica. The small percentages of chlorine are due to the HCl used for the hydrolysis of tetraethoxysilane, whereas the phosphorus has its origin in the buffering reagents.

#### 3.4.2. Size (DLS) and Charge (ζ-Potential)

To get a first impression of the effects of uranyl cation adsorption, two additional experiments were designed. The size and charge of the xerogel aggregates adsorbing in a gradually more concentrated uranyl cation solution have been measured at pH 3. As is naturally expected, xerogel aggregates became bigger when exposed to gradually higher uranium concentrations (Figure 9a). Furthermore, in conformity with the behavior of the polyamide microplastics adsorbing europium and uranium [132], a decrease in the charge that develops at the interface between the liquid and the solvation sphere surface by increasing uranium concentration has been observed (Figure 9b). This decrease, though, is less intense, and the hybrid adsorbents maintain a positive charge throughout the entire concentration range. This phenomenally unexpected tendency has been attributed to the development of an anion solvation sphere with hydroxyl and nitrate anions that counterbalance the positive charge of the adsorbed uranyl cations. The observed considerable increase in hydrodynamic radii is thus not attributed solely to the accumulation of uranyl cations but also to the formation of larger solvation spheres. Analogous behavior has been encountered during the investigation of the size and charge of the silica nanoparticles adsorbing at different pHs (Figure 9c). The increase in the hydrodynamic radius indicates a higher adsorption capacity and accumulation of uranyl cations at pH 3, as was verified for nanoparticles 75,000 (compare Figure 5a with Figure 5b) and, in parallel, a larger solvation sphere as discussed above. The “unanticipated” lower charge of the nanoparticles at pH 3 (Figure 9d) corroborates the theory of the larger concentration of counter anions in their solvation sphere.

#### 3.4.3. IR Spectroscopy

The formation of inner-sphere complexes between uranium and the composites with the hyperbranched poly(ethylene imine) matrix is indicated by the associated FTIR spectra, which have been obtained after U(VI) adsorption at different initial concentrations and are shown in Figure 10. The spectra show significant changes in the structure and intensity ratio of the absorption bands related to the surface-active groups of the dendritic polymers (peaks in the range of 1700 and 1000 cm^−1^) after interaction with U(VI) [93]. These changes are related to the formation of inner-sphere complexes between U(VI) and the surface-active groups, which affect the geometry and bonding strength of the interacting groups. There is also a significant shift of the silanol stretching bands from 960 to 900 cm^−1^. Moreover, the peak, which appears at 1010 cm^−1^ and 970 cm^−1^ in the FTIR spectra of the nanoparticles and xerogels, respectively, is attributed to the asymmetric band (ν3) of the [O=U=O]^2+^ moiety [133], and its intensity increases with the amount of U(VI) adsorbed, indicating the progressing and supreme U(VI) adsorption by the dendrimers. Moreover, the sharp peak at 1380 cm^−1^ is ascribed to nitrate anions (NO_3_^−^) [134], which act as counter-ions for the U(VI) cationic surface complexes. It is very interesting to note in all cases the profound escalation in the intensity of this NO_3_^−^ band by increasing uranium concentration. This is another piece of evidence that supports the accumulation of nitrates in the solvation sphere and explains the previously discussed ζ-potential decrease. It further indicates that the anions are adsorbed into the composite surface and remain in the dried samples.

## 4. Conclusions

Dendritic PEI proved to be an ideal matrix for biomimetic silicification. Furthermore, besides the beneficial fusion of an organic and an inorganic adsorbent, an adequate alternative to the tedious processes required for the chemical attachment of a dendritic polymer to silica particles was provided. The novelty introduced by this work is that a simple variation of the orthosilicic acid/hyperbranched poly(ethylene imine) ratio yields two materials with different properties and capabilities: silica hydrogels-xerogels and precipitated silica nanoparticles. Both can be used in the form of dispersed powders and are potent adsorbing materials. Gelation, in contrast to precipitation, may take place on porous substrates. This implementation of xerogels as coatings offers many more possibilities, such as enhancement of the pollutant removal potential of an already porous adsorbent and immobilization on an appropriate scaffold for easy removal and reuse. On the other hand, precipitated nanoparticles contain double quantities of PEI. This larger organic content enhances their capacity, which is among the best in the literature. Hybrid silica dendritic polymer composites may additionally be regenerated by changing the pH or by washing organic solvents with mild heating [135,136,137,138]. The performance of these hybrid materials equals or even surpasses that of their conventionally synthesized counterparts. They are far cheaper and more environmentally friendly since their production does not require high temperatures or toxic organic solvents. Their most promising property, though, is the ability of PEI to combine silicification with biomimetic mineralization [139]. The transformation of noble metal ions into nanoparticles allows synchronous adsorption and (photo)catalytic decomposition of many toxic pollutants [140]. Both composites may benefit from this possibility, and an investigation is currently underway with promising results.

## Figures and Tables

**Figure 1 nanomaterials-13-01794-f001:**
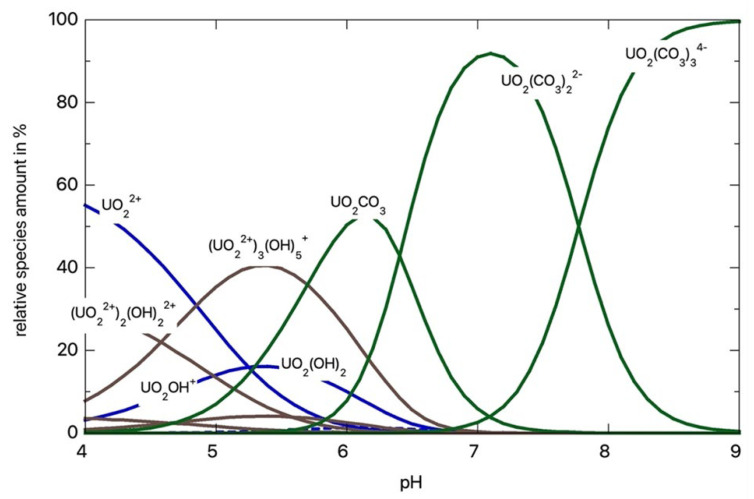
Uranium species distribution percentages as a function of pH.

**Figure 2 nanomaterials-13-01794-f002:**
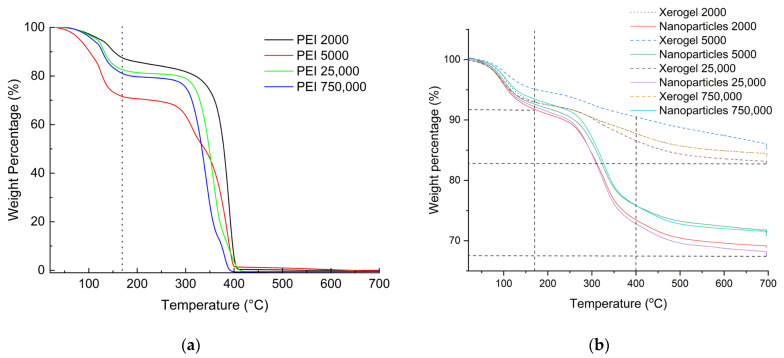
Thermogravimetry profiles for the dendritic matrices (**a**) (Adapted with permission from Ref. [97] 2022 Elsevier) and the hybrid adsorbents (**b**).

**Figure 3 nanomaterials-13-01794-f003:**
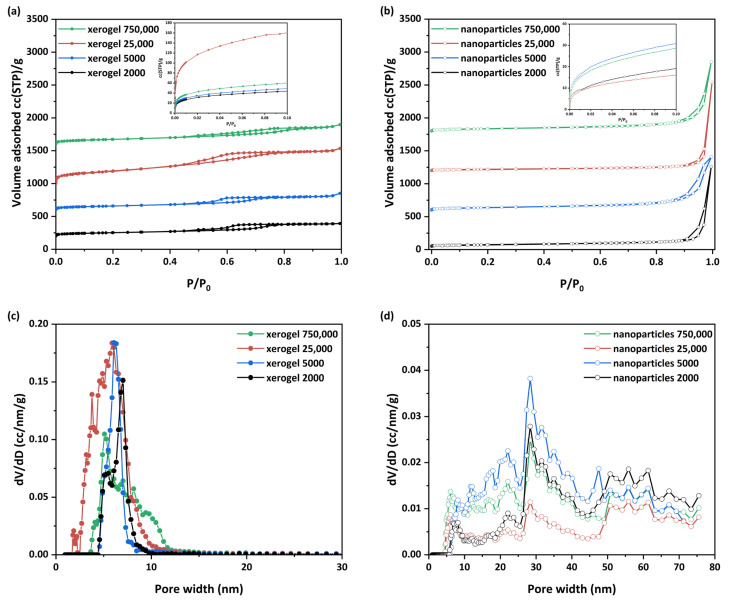
N2 adsorption (77 K) of the prepared (**a**) xerogels and (**b**) nanoparticles; in insets, N2 adsorption (77 K) of all samples in the low-pressure region is depicted. Pore size distributions determined by the NLDFT method of (**c**) xerogels and (**d**) nanospheres.

**Figure 4 nanomaterials-13-01794-f004:**
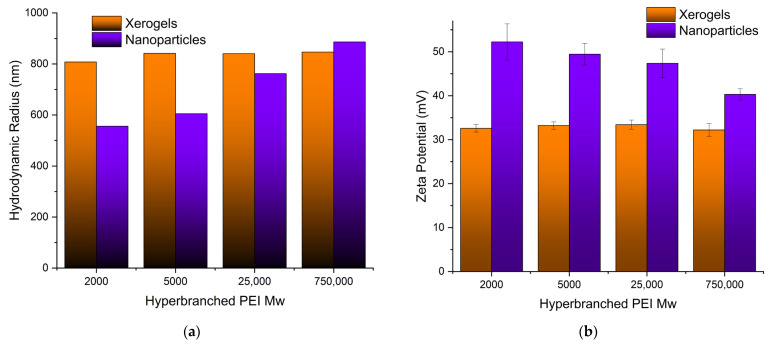
(**a**) Hydrodynamic radii and (**b**) surface charges of all adsorbing materials as measured by dynamic light scattering and ζ-potential, respectively.

**Figure 5 nanomaterials-13-01794-f005:**
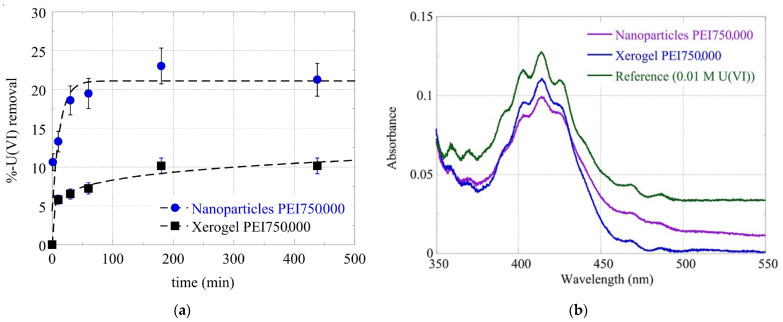
(**a**) Removal percentage of uranyl cations from nanoparticles and xerogel Mw 750,000 as a function of time and respective absorbance curves after 2 h (**b**).

**Figure 6 nanomaterials-13-01794-f006:**
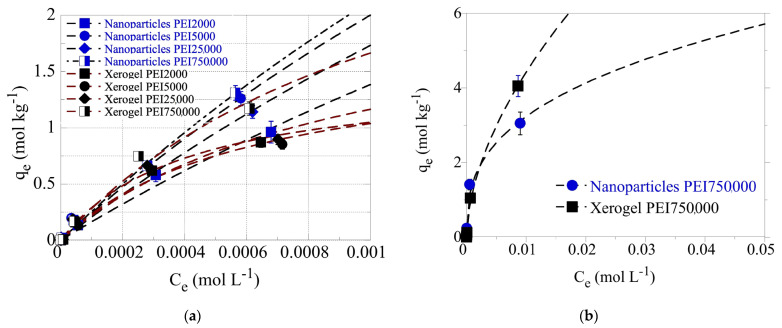
(**a**) Isotherms of U(VI) adsorbed by xerogel and nanoparticle adsorbents at 283 K and varying initial U(VI) levels in solution (1 × 10^−5^–0.1 mol/L). The experiments were performed at pH 4 (left) (**a**) and pH 3 (**b**), at an adsorbent dosage of 0.4 g/L, under ambient conditions (298 K), 24 h contact time, and an agitation rate of 125 rpm.

**Figure 7 nanomaterials-13-01794-f007:**
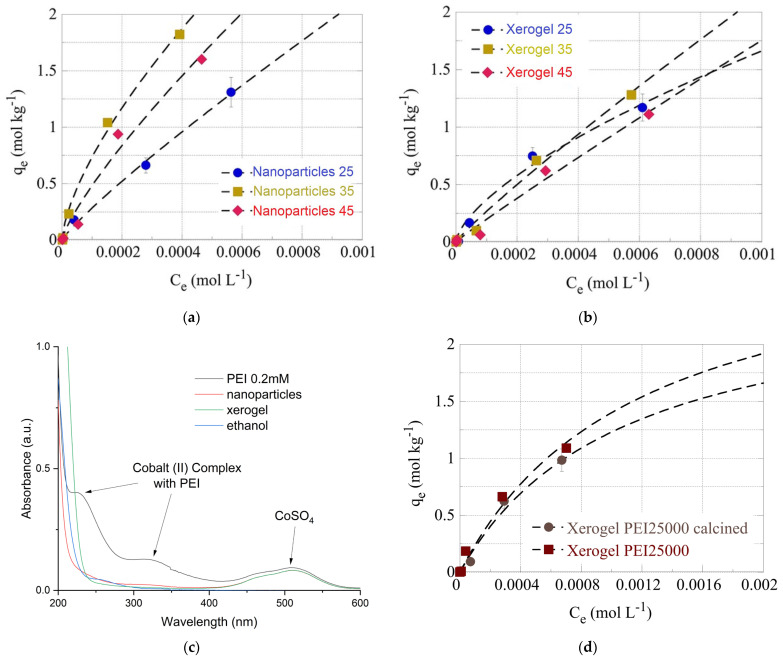
Isotherms of U(VI) adsorbed by PEI 750,000 nanoparticles (**a**) and xerogel (**b**) adsorbents at various temperatures (298 K, 308 K, and 318 K). (**c**) UV-visible spectra of CoSO_4_ solution, respective complex with PEI, supernatants of 100 mg xerogel, and nanoparticles after 1 week at 45 °C and ethanol. (**d**) Isotherms of U(VI) adsorbed by calcinated and non-calcinated PEI 25,000 nanoparticles.

**Figure 8 nanomaterials-13-01794-f008:**
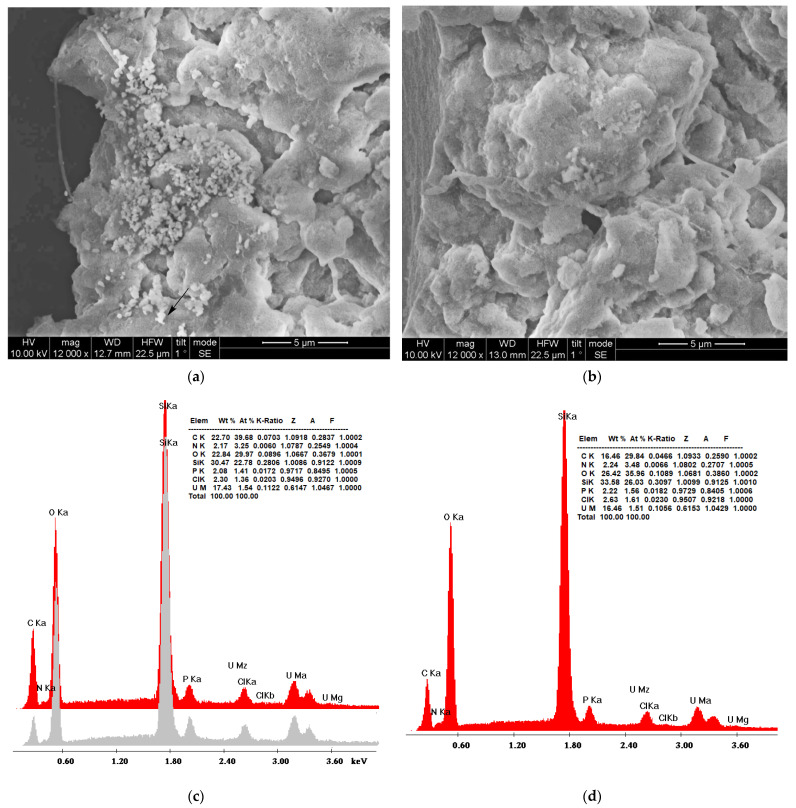
SEM micrographs of composite PEI-silica nanoparticles (**a**) and silica xerogels (**b**) and respective energy dispersive spectroscopy (EDS) diagrams (**c**,**d**) after uranyl cation adsorption. The gray diagram indicates the EDS spectrum in the bright spot indicated by the arrow.

**Figure 9 nanomaterials-13-01794-f009:**
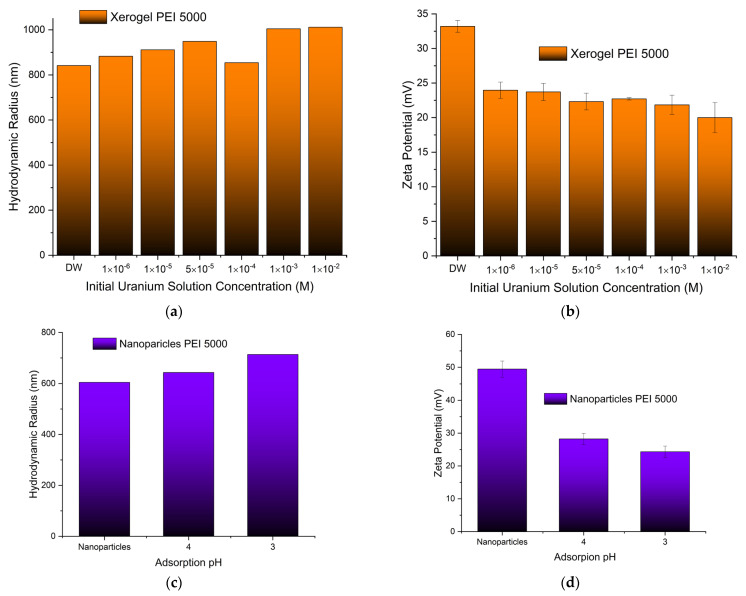
Hydrodynamic radii (**a**) and surface charge (**b**) of xerogels adsorbing uranyl cations as a function of the initial solution concentration. Hydrodynamic radii (**c**) and surface charge (**d**) of nanoparticles adsorbing uranyl cations as a function of the solution pH.

**Figure 10 nanomaterials-13-01794-f010:**
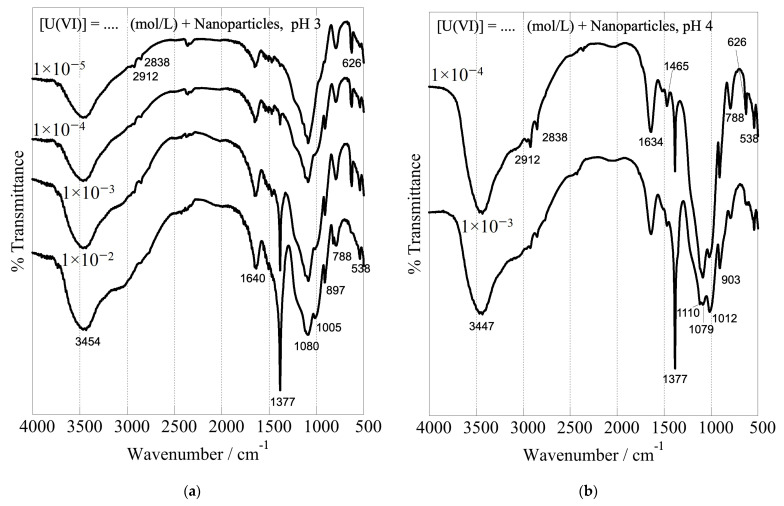
FTIR spectra of nanoparticles at pH 3 (**a**) and pH 4 (**b**) and xerogels at pH 3 (**c**) after U(VI) adsorption at varying initial U(VI) levels in the suspension. The associated experiments were performed in a U(VI) initial concentration range between 1 × 10^−5^ and 1 × 10^−2^ mol L^−1^, with an adsorbent dosage of 0.4 g/L under ambient conditions (298 K), a 24 h contact time, and an agitation rate of 125 rpm.

**Table 1 nanomaterials-13-01794-t001:** Weight loss percentages of the hybrid adsorbents.

Adsorbent	Weight Loss 170 °C (Water/Ethanol) (%)	Total Weight Loss 3 h 700 °C (%)	Organic Content (%)
Xerogel 2000	6.99	15.97	8.98
Xerogel 5000	4.92	14.89	9.91
Xerogel 25,000	7.15	17.28	10.13
Xerogel 750,000	7.05	16.01	8.96
Nanoparticles 2000	8.28	31.08	22.80
Nanoparticles 5000	7.40	28.90	21.50
Nanoparticles 25,000	7.80	32.23	24.43
Nanoparticles 750,000	6.53	28.84	22.31

**Table 2 nanomaterials-13-01794-t002:** FTIR vibrational band assignments (in cm^−1^) for hyperbranched poly(ethylene imine) and composite silica xerogels and nanospheres (nanoparticles).

Band Assignment ^1^	PEI25000	Xerogels	Nanoparticles
*ν_s_* SiO-H free	-	3750 (vw)	-
*ν_s_* SiO-H Hydrogen bonded	-	3450 (*w*/*b*)	3450 (*w*/*b*)
*ν_as_* NH (primary, secondary)	3350 (m)	3400 (w)	3370 (sh)
*ν_s_* NH (primary, secondary)	3276 (m)	op	3200 (w)
*ν_as_* CH_2_	2935 (m)	2981 (vw)	2962 (vw)
*ν_s_* CH_2_	2810 (s)	2820 (vw)	2853 (vw)
*δ* NH, NH_2_	1585 (m)	1640 (vw)	-
*δ_as_* NH_2_^+^, NH_3_^+^	-	1530 (vw)	1524 (vw)
*δ_s_* NH_2_^+^, NH_3_^+^	-	-	1473 (vw)
*ν_as_* C-N	1105 (m)	sh	sh
*ν_s_* C-N	1045 (m)	op	op
*ν* Si-O-Si	-	1074 (s)	1042 (s)
*ν* Si-OH, Si-O^-^	-	960 (m)	964 (m)
*δ* Si-O-Si	-	788 (m)	785 (m)
*ρ* CH_2_	760 (s)	op	op
*δ Si-OH*	-	540 (m)	540 (m)

^1^ Assignments: *ν* (stretch), *δ* (bend), and *ρ* (rock). The subscripts as and s denote asymmetric and symmetric vibrations, respectively. Band intensities: s (strong), m (medium), w (weak), vw (very weak), sh (shoulder), br (broad), and op (overlapped).

**Table 3 nanomaterials-13-01794-t003:** BET surface areas and pore characteristics of all samples.

Sample	TPV ^1^	S_BET_	d_mean_ ^2^	d_BJH_ ^3^	d_DFT_ ^4^
	(mL/g)	(m^2^/g)	(nm)	(nm)	(nm)
Nanoparticles 750,000	0.640	139.0	18.4	32.0	28.4
Nanoparticles 25,000	0.326	77.9	16.7	53.5	55.8
Nanoparticles 5000	0.880	148.3	23.7	31.0	28.4
Nanoparticles 2000	0.508	98.5	20.6	53.6	28.4
Xerogel 750,000	0.461	271.4	6.8	3.8	5.1
Xerogel 25,000	0.829	714.1	4.6	3.8	5.9
Xerogel 5000	0.390	220.3	7.1	4.7	6.1
Xerogel 2000	0.297	194.3	6.1	5.4	7.0

^1^ Total pore volume at 0.99. ^2^ Mean pore size d_mean_ as 4000·TPV/S_BET_. ^3^ Pore size was determined from the pore size distribution (PSD) using the Barrett-Joyner-Halenda (BJH) method based on the Kelvin equation of the N_2_ desorption branch. ^4^ Pore diameter was determined from the pore size distribution (PSD) curve based on the DFT model (for the Non-Local Density Functional Theory (NLDFT) method, an equilibrium kernel for silica as an adsorbent and N_2_ 77 K as an adsorbate was used).

**Table 4 nanomaterials-13-01794-t004:** Specific surface areas of silica nanoparticles/xerogels and relevant adsorbents.

Sample	S_BET_ (m^2^/g)	Reference
Nanoparticles 5000	148.3	This Work
Xerogel 25,000	714.1	This Work
Nano-silica composites- PEI 30%	104	[111]
Nano-silica composites- PEI 60%	0.65	[111]
Mesoporous silica microcapsules	514	[112]
Silica xerogels	363.72	[113]
Silica xerogels	335–850	[114]
PEG-silica xerogels	19–127	[115]

**Table 5 nanomaterials-13-01794-t005:** Adsorption capacities of the nanoparticle and xerogel samples in comparison with other adsorbents.

Sample	q_max_ (mol kg^−1^)	Reference
Nanoparticles 750,000	11.5	This Work
Nanoparticles 25,000	9.5	This Work
Nanoparticles 5000	10.6	This Work
Nanoparticles 2000	8.1	This Work
Xerogel 750,000	3.7	This Work
Xerogel 25,000	1.5	This Work
Xerogel 5000	1.7	This Work
Xerogel 2000	2.3	This Work
Nano-silica with hyperbranched PAMAM	0.91	[124]
Hydroxyapatite with konjac gum	9.0	[125,126]
Polyurea-crosslinked alginate	8.72	[127]
Reduced graphene oxide/ZIF-67	8.14	[128]
Al_2_O_3_/MgO	4.51	[129]
MOF/black phosphorus quantum dots on cellulose	3.7	[130]

## Data Availability

There are no data available.

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
