# Peer review of "Comparative Study of the U(VI) Adsorption by Hybrid Silica-Hyperbranched Poly(ethylene imine) Nanoparticles and Xerogels"

_nanomaterials, 2023, doi:10.3390/nano13111794_

Round 1

Reviewer 1 Report

Dear Authors,

The paper concerns comparative studies of U(VI) adsorption on two types of adsorbents. In my opinion, the subject of the work is exciting. The concept of work is understandable. Many experiments were carried out in the work, mainly well described. Nevertheless, some parts should be extended or explained in more detail.

Line 41-49

The Authors wrote that uranium exhibits a very complex behavior in water. A few sentences below, the authors note that the predominant form of uranium compounds in water is, i.e., the hexavalent uranyl cation (UO22+). Maybe it's worth introducing a drawing with the predicted speciation (% distribution) of U as a function of pH (e.g., as in references [1, 2].

Line 139

The pH values shown in line 139 are the initial values. How were solutions with this pH obtained? Were the pH values of the solutions after adsorption measured? Was a buffer used?

Line 150

Adsorption isotherms Langmuir and Freundlich describe how molecules are distributed between an adsorbent and an adsorbate when the adsorption process has come into equilibrium. The authors describe the Langmuir and Freundlich isotherms in the kinetics of adsorption chapter. In my opinion, adsorption kinetics are described by pseudo-first and pseudo-second order equations.

Therefore, they should be described in a separate subsection of methods, and the calculated results can be shown in Chapter 3.2.

line 299

How do you know the adsorption time of mercury and cadmium?

Line 322

Only one value for qmax is given in section 3.3. (line 332). It was probably calculated, but what equation was used? I suggest introducing a table with adsorption isotherms parameters calculated by Langmuir and Freundlich adsorption (e.g., ???? (mg/g), KL(L/mg) ....).

Linia 323

Maybe it's worth inserting a table with the adsorbents mentioned by the authors and their ?????

Author Response

Reviewer 1

The paper concerns comparative studies of U(VI) adsorption on two types of adsorbents. In my opinion, the subject of the work is exciting. The concept of work is understandable. Many experiments were carried out in the work, mainly well described. Nevertheless, some parts should be extended or explained in more detail.

Thank you very much for the polite and very encouraging comments

Line 41-49

The Authors wrote that uranium exhibits a very complex behaviour in water. A few sentences below, the authors note that the predominant form of uranium compounds in water is, i.e., the hexavalent uranyl cation (UO22+). Maybe it's worth introducing a drawing with the predicted speciation (% distribution) of U as a function of pH (e.g., as in references [1, 2].

A new figure (Figure 1) is inserted in the revised manuscript and depicts uranium behaviour in different pHs.

Line 139

The pH values shown in line 139 are the initial values. How were solutions with this pH obtained? Were the pH values of the solutions after adsorption measured? Was a buffer used?

The pH values correspond to the pH values determined at equilibrium after adsorption. There was no buffer needed to stabilize the pH because the desired pH was adjusted using 0.1 HCl or 0.1 M NaOH and remained stable (pH 3 and pH 4) after adjustment. This is now incorporated into the text  

Line 150

Adsorption isotherms Langmuir and Freundlich describe how molecules are distributed between an adsorbent and an adsorbate when the adsorption process has come into equilibrium. The authors describe the Langmuir and Freundlich isotherms in the kinetics of adsorption chapter. In my opinion, adsorption kinetics are described by pseudo-first and pseudo-second order equations.

Therefore, they should be described in a separate subsection of methods, and the calculated results can be shown in Chapter 3.2.

The adsorption isotherms methods are now described in a separate subsection of methods in Chapter 2.4. The results are discussed in Chapter 3.3. The associated data have been fitted by the Langmuir isotherm model. 

line 299

How do you know the adsorption time of mercury and cadmium?

It is the same reference as the dichromates. Now it is included in the text. 

Line 322

Only one value for qmax is given in section 3.3. (line 332). It was probably calculated, but what equation was used? I suggest introducing a table with adsorption isotherms parameters calculated by Langmuir and Freundlich adsorption (e.g., ???? (mg/g), KL(L/mg) ....).

We fitted the adsorption data used by the Langmuir isotherm model just to evaluate the qmax value. Because the isotherm models are empirical and have been developed for gas adsorption, the use of the KL value and other parameters to describe mechanisms in complex aqueous systems is very limited.

Line 323

Maybe it's worth inserting a table with the adsorbents mentioned by the authors and their ?????

A new table (Table 4 in the revised version) is inserted in the manuscript containing the qmax values of the adsorbents together with characteristic values of dendritic fibrous nano-silica with hyperbranched poly(amidoamine) and aerogel counterparts

Reviewer 2 Report

The manuscript deals to produce and characterize two different silica conformations (Xerogels and Nanoparticles), and U(VI) adsorption was investigated. The subject is interesting and up-to date, it is related to the profile of the journal; however, the novelty of the content must be emphasized. The structure of the manuscript is generally clear, several up-to date methods were used. The English of the manuscript is good, few spelling mistakes can be found, please check it carefully.

Comments and questions:

  1. The Title is informative.
  2. The Abstract reflects the approach of the study, it summarizes the findings of the work, however, the application of the adsorbent also should be indicated in the abstract.
  3. The section Introduction presents the important points of the topic, it contains references related to the earlier results, reveals the importance and originality of the work.
  4. Materials and methods: The experimental design is appropriate and adequately described.
  5. Results and discussion: In this section, authors describe the results shown in the corresponding figures, and tables. The figures generally are nice, but in some cases the legends or lines could be more informative:

·         Fig 1. It would be more easy to understand the figure if the different type of adsorbents would be symbolized with different line style, (e.g. xerogels by dashed line, nanoparticles by solid lines..)

·         Fig 2. c. It was explained in the text that xerogel 25000 was measured in different conditions; however as the figures must be self-explaining, please indicate this (e.g. give the pressure) in the figure title.

·         Fig 4a. and Fig 5, The colors of different adsorbents are very similar, even in colored monitor. Please consider changing the color of one of the adsorbents to be distinguishable even is black and white photocopies.

·         Generally, for established isotherm fitting more than 3 measured points should be involved into the calculations.

·         Fig 9. Please indicate the concentrations in the figures, moreover, please explain what the abbreviations 1E-2 ect. mean.

  1. The Conclusions section summarizes shortly the result of the work.
  2. Conclusion and recommendation: This manuscript is recommended for publication after minor revision.

In some cases splelling mistakes were detected (e.g. period at the end of the sentences)

Author Response

Reviewer 2

The manuscript deals to produce and characterize two different silica conformations (Xerogels and Nanoparticles), and U(VI) adsorption was investigated. The subject is interesting and up-to date, it is related to the profile of the journal; however, the novelty of the content must be emphasized. The structure of the manuscript is generally clear, several up-to date methods were used. The English of the manuscript is good, few spelling mistakes can be found, please check it carefully.

 Thank you very much for the polite and very encouraging comments

Comments and questions:

  1. The Title is informative.
  2. The Abstract reflects the approach of the study, it summarizes the findings of the work, however, the application of the adsorbent also should be indicated in the abstract.

Two applications of the adsorbents are now included in the abstract

  1. The section Introduction presents the important points of the topic, it contains references related to the earlier results, reveals the importance and originality of the work.
  2. Materials and methods: The experimental design is appropriate and adequately described.
  3. Results and discussion: In this sectionauthors describe the results shown in the corresponding figures, and tables. The figures generally are nice, but in some cases the legends or lines could be more informative:
  • Fig 1. It would be more easy to understand the figure if the different type of adsorbents would be symbolized with different line style, (e.g. xerogels by dashed line, nanoparticles by solid lines..)

Xerogels are now symbolized in dashed lines (we kept Xerogel 2000 depicted in dots to avoid an optical illusion giving the impression there are 3 curves instead of four)

  • Fig 2. c. It was explained in the text that xerogel 25000 was measured in different conditions; however as the figures must be self-explaining, please indicate this (e.g. give the pressure) in the figure title.

All measurements have been performed in the same conditions (i.e. same pressure range). The differentiation of the xerogel 25000 is a slightly wider hysteresis loop in the area of the higher relative pressures. This is depicted in Figure 2a. This is now clarified in the text.

  • Fig 4a. and Fig 5, The colors of different adsorbents are very similar, even in colored monitor. Please consider changing the color of one of the adsorbents to be distinguishable even is black and white photocopies.

We have changed the colour of the “nanoparticles” adsorbent to blue.

  • Generally, for established isotherm fitting more than 3 measured points should be involved into the calculations.

Generally, the experimental data involved in the calculations (fits) are more than five. Some data points in the lower Ce are merely overlapped.

  • Fig 9. Please indicate the concentrations in the figures, moreover, please explain what the abbreviations 1E-2 ect. mean.

1-E-2 stands for 1×102 mol L-1, 1-E-3 stands for 1×103 mol L-1 etc. A small addition has been inserted in the legend and now it is self-explanatory

  1. The Conclusions section summarizes shortly the result of the work.
  2. Conclusion and recommendation: This manuscript is recommended for publication after minor revision.

Reviewer 3 Report

The paper focus on the purification of water by using two different silica conformations (Xerogels and Nanoparticles). The author also investigated the effect of crucial factors i.e., temperature, electrostatic forces, adsorbent composition….

 In general, the results mostly support the authors' conclusions. However, some aspects of the manuscript must be carefully reviewed, discussed and improved.

1°) The originality, mechanism, and scientific reliability of the reactor configuration are unclear. In my opinion, there are some major points that the authors should address before it is accepted for publication. Please insist on NOVELTY (type of adsorbent?)

2°) page 2: Authors are invited to add more references about Adsorbents (low-cost) , I suggest adding more references to manuscript : Journal of cleaner production 201, 28-38 (2018); Environmental Science and Pollution Research 30, 39169–39183 (2023)

3°) From table 3, it seems that the specific surface does not exceed 300 mg/g for each adsorbent. Authors should compare their results with others from the literature

4°) Figure 4.a : Please add standard bars

5°) the part about Aadsrobent reusability is not provided. For sake clarity, explanations are needed.

Author Response

Reviewer 3

The paper focus on the purification of water by using two different silica conformations (Xerogels and Nanoparticles). The author also investigated the effect of crucial factors i.e., temperature, electrostatic forces, adsorbent composition….

 In general, the results mostly support the authors' conclusions. However, some aspects of the manuscript must be carefully reviewed, discussed and improved.

1°) The originality, mechanism, and scientific reliability of the reactor configuration are unclear. In my opinion, there are some major points that the authors should address before it is accepted for publication. Please insist on NOVELTY (type of adsorbent?)

The novelty of the adsorbents is now highlighted in the conclusion section

2°) page 2: Authors are invited to add more references about Adsorbents (low-cost) , I suggest adding more references to manuscript : Journal of cleaner production 201, 28-38 (2018); Environmental Science and Pollution Research 30, 39169–39183 (2023)

More references about adsorbents are now incorporated in the manuscript including those proposed by the reviewer

3°) From table 3, it seems that the specific surface does not exceed 300 mg/g for each adsorbent. Authors should compare their results with others from the literature

A new table containing specific surface data of similar adsorbents has been inserted in the manuscript

4°) Figure 4.a : Please add standard bars

Standard bars are now added in Figure 4a

5°) the part about Adsrobent reusability is not provided. For sake clarity, explanations are needed.

The regeneration of hybrid silica-dendritic polymers is feasible. Relevant citations are now included in the manuscript

Round 2

Reviewer 3 Report

Authors have addressed all my points. The Ms has improved a lot, very interesting paper actually. I can recommend the Ms for publication now.